# Practical Entropy Accumulation for Random Number Generators with Image Sensor-Based Quantum Noise Sources

**DOI:** 10.3390/e25071056

**Published:** 2023-07-13

**Authors:** Youngrak Choi, Yongjin Yeom, Ju-Sung Kang

**Affiliations:** 1Department of Financial Information Security, Kookmin University, Seoul 02707, Republic of Korea; alpha1996@naver.com; 2Department of Mathematics, Kookmin University, Seoul 02707, Republic of Korea; salt@kookmin.ac.kr

**Keywords:** entropy accumulation, random number generator, quantum random noises

## Abstract

The efficient generation of high-quality random numbers is essential in the operation of cryptographic modules. The quality of a random number generator is evaluated by the min-entropy of its entropy source. The typical method used to achieve high min-entropy of the output sequence is an entropy accumulation based on a hash function. This is grounded in the famous Leftover Hash Lemma, which guarantees a lower bound on the min-entropy of the output sequence. However, the hash function-based entropy accumulation has slow speed in general. For a practical perspective, we need a new efficient entropy accumulation with the theoretical background for the min-entropy of the output sequence. In this work, we obtain the theoretical bound for the min-entropy of the output random sequence through the very efficient entropy accumulation using only bitwise XOR operations, where the input sequences from the entropy source are independent. Moreover, we examine our theoretical results by applying them to the quantum random number generator that uses dark shot noise arising from image sensor pixels as its entropy source.

## 1. Introduction

A random number generator (RNG) is an important component of cryptographic systems used by cryptographic modules to generate random numbers. Random numbers are used for various purposes including the generation of cryptographic keys, and their significance has been increasingly highlighted, particularly with the recent emergence of quantum key distribution [1].

An RNG can be divided into three main processes: digitization, entropy accumulation, and pseudo random number generation (PRNG). Digitization is the process of converting entropy sources into binary data. We call the converted binary data the “input sequence”. Typically, the input sequence has a low min-entropy. Entropy accumulation is the process of transforming input sequences into data with high min-entropy. We denote the input sequence that has undergone the entropy accumulation process as the “output sequence”. PRNG is composed of deterministic algorithms, such as block ciphers or hash functions, and it assumes the output sequence as input and then outputs the final “random number”. The operation of the RNG is illustrated in Figure 1.

Although unpredictable random numbers can be generated using a digitized entropy source, this is impractical in cryptographic systems because of the significant amount of time required. The PRNG was used to address this limitation. A PRNG produces the same output with the same input. This implies that the generated random numbers are not unpredictable. However, the PRNG can generate multiple random numbers in a short time because the length of the output is longer than the length of the input. Therefore, if the length of the input of the PRNG is small but random, the output of the PRNG provides good random numbers. Consequently, the high min-entropy of the input sequence can be observed as an important factor in constructing an RNG.

In the process of entropy accumulation, an accumulation function *H* is a transformation from an input sequence to an output sequence. Let *X* be an input sequence and $X^{\prime }$ be the corresponding output sequence, then the entropy accumulation can be expressed as $X^{\prime }=H\left(X\right)$.

On the other hand, Dodis et al. [2] have proposed that the entropy accumulation is divided into two types depending on the characteristics of the accumulation function *H*. The first type is called “Slow-Refresh”, which is characterized by a high computational complexity of *H* leading to slower accumulation speed but results in a long output sequence. The second type, so called “Fast-Refresh”, features a lower computational complexity of *H* leading to faster accumulation speed but results in a short output sequence.

Traditional hash function-based entropy accumulation is categorized as Slow-Refresh due to its relatively slow accumulation speed. However, this method is widely utilized because of the theoretical foundation provided by the Leftover Hash Lemma [3]. The Leftover Hash Lemma ensures the lower bound of min-entropy for the output sequence, where a low min-entropy input sequence passes through a hash function *H* randomly selected from a universal hash family.

There are two major considerations in the Slow-Refresh process, which are the construction of a universal hash family and the random selection of a hash function from the uniformly distributed universal hash family. However, constructing a universal hash family is not a trivial task. One example satisfying an appropriate property is the family of the Hankel matrix [4]. If a Hankel matrix is randomly selected from the uniformly distributed family, and an *n*-bit input sequence is processed through this selected matrix, then the resulting *m*-bit output sequence will have a sufficient min-entropy. Similarly, a universal hash family can also be constructed by using Toeplitz matrices [5,6].

In such a matrix-based universal hash family, the matrices typically used have input bit-lengths larger than the output bit-lengths, and the ratio m/n approaches 1 as *n* increases. Thus, the size of the matrix must be sufficiently large in order to minimize the lost bits. However, this leads to a high computational complexity, that is, this method falls under the Slow-Refresh category, and Slow-Refresh may not always be suitable in practical situations that require rapid entropy accumulation. Therefore, in this work, we focus on Fast-Refresh, which is clearly described in the next subsection.

### 1.1. Related Works

One of the typical examples of Fast-Refresh without a hash function is Microsoft Windows RNG [7]. Windows RNG uses only the bitwise XOR operation and bit permutation $rot_{\left(\alpha ,n\right)}$ for the entropy accumulation. In particular, if we employ the following notation, the entropy accumulation operation of Windows RNG can be depicted as shown in Figure 2.

$Y_{1},Y_{2},\cdots ,Y_{l}$: *n*-bit input sequences.π:{0,1,⋯,n−1}→{0,1,⋯,n−1}, π is one-to-one.$A_{\pi }:{\left\{0,1\right\}}^{n}\to {\left\{0,1\right\}}^{n}$, $A_{\pi }\left(b_{0},b_{1},\cdots ,b_{n-1}\right):=A_{\pi }\left(b_{\pi (0)},b_{\pi (1)},\cdots ,b_{\pi (n-1)}\right)$.$rot_{\left(\alpha ,n\right)}:\left\{0,1,\cdots ,n-1\right\}\to \left\{0,1,\cdots ,n-1\right\}$, $rot_{\left(\alpha ,n\right)}\left(i\right):=i-\alpha (\;\;\;\;\;\mathrm{mod}\;\;n$).

Figure 3 is an example of the $rot_{\left(3,8\right)}$ operation on an 8-bit input sequence.

$\Lambda_{\pi }^{\left(l\right)}:{\left\{0,1\right\}}^{n}\to {\left\{0,1\right\}}^{n}$, $\Lambda_{\pi }^{\left(l\right)}:=\left\{\begin{array}{cc} Y_{1} & \mathrm{if}\;l=1\\ A_{\pi }\left(\Lambda_{\pi }^{\left(l-1\right)}\right)⊕Y_{l} & \mathrm{if}\;l\geq 2 \end{array}\right.$

Windows RNG accumulates output sequences relatively quickly in an entropy pool because it uses bit permutations and bitwise XOR operations without employing hash functions. Despite these advantages, it has not been proven whether the entropy accumulation of Windows RNG guarantees a lower bound for the min-entropy of the output sequence, as does the Leftover Hash Lemma when using a hash function. Therefore, it has been challenging to consider this method of secure entropy accumulation. However, recent research presented at Crypto 2021 analyzed Microsoft Windows RNG. In [2], the security of Windows RNG was analyzed by providing the number of iterations of bit permutation and bitwise XOR to surpass an arbitrary min-entropy under three conditions. First, the input sequences must be independent. Second, the probability distribution of input sequences must follow the “2-monotone distribution”. Third, the “covering number” of the bit permutation must be finite. In [2], it was claimed that the three conditions just mentioned are easy to satisfy. However, satisfying these conditions may be challenging for hardware entropy sources rather than software entropy sources, particularly when managing multiple entropy sources. The first and third conditions are easily satisfied, as in the case of Windows RNG. The second condition may seem easy to achieve, but it is challenging. Therefore, to handle entropy sources other than Windows, a more relaxed condition is required than that presented in [2].

### 1.2. Our Contributions

The contributions of this paper can be summarized into three main aspects. First, we provide Fast-Refresh that does not require hash functions and uses only bitwise XOR operations to generate output sequences. In particular, if we employ the following notation, the proposed entropy accumulation can be depicted as shown in Figure 4.

$Y_{1},Y_{2},\cdots ,Y_{l}$: *n*-bit input sequences.$\Gamma^{\left(l\right)}=\sum_{j=1}^{l}Y_{j}$.

This method requires only two conditions for the input sequences, making it relatively easier to satisfy than Windows RNG entropy accumulation.

Second, we establish a min-entropy lower bound for a secure random number generator and demonstrate that our entropy accumulation successfully surpasses this lower bound when applied to our RNG. We used quantum random number generators (QRNGs) as an entropy source. QRNGs utilize several quantum phenomena to generate high-quality random numbers [8,9,10].

The first published QRNG was based on radioactive decay. This emerged as the need for random number generators increased alongside the rise of computer simulations in the late 20th century. QRNGs with radioactive decay utilize the random behavior of particles emitted from radioactive materials. This method is still in use today, and, along with methods using the photon, it is one of the most common QRNG [8]. There are various quantum phenomena used for quantum random number generation. For example, the study in [9] uses the interference of photons to generate random numbers. Another example from [11] involves the use of tunneling signals in silicon diodes for random number generation. Furthermore, the work in [12] employs short laser pulses with quantum random phases to generate random numbers.

In this paper, we use the image sensor-based QRNG of [13]. The image sensor-based QRNG generates random numbers using dark shot noise. Dark shot noise is a fluctuation of small current that flows through the pixels of the image sensor even when they do not receive light. It is known that the number of electrons follows the Poisson distribution [14,15]. For this reason, we utilize optical black pixels (OBP) of the image sensor, which do not receive light, as the entropy source. Furthermore, because each pixel outputs dark shot noise independently, all the entropy sources can be considered independent of each other.

Third, we conduct a comparative analysis of our proposed entropy accumulation with other entropy accumulations. The examples chosen for comparison are the Slow-Refresh used in IDQ QRNG and the Fast-Refresh of Windows RNG. When comparing with the Slow-Refresh, we focus on the theoretical differences in the accumulation mechanisms between ours and the IDQ QRNG. Under the view point of Fast-Refresh, we compare efficiency of ours with Windows RNG by evaluating the iteration number of operations.

The remainder of this paper is organized as follows. In Section 2, we describe the theoretical background and propose our main theorem, which guarantees the lower bound of min-entropy of the output sequence. In Section 3, we describe the process of applying the theory outlined in Section 2 to an image sensor-based QRNG. We establish a min-entropy lower bound based on three standards and provide experimental results demonstrating that the output sequences generated by applying our theory to the input sequences have a min-entropy higher than the established lower bound. In Section 4, we compare our entropy accumulation with other entropy accumulations. The first comparison is with Slow-Refresh of IDQ QRNG. We describe the theoretical background of Slow-Refresh and the Leftover Hash Lemma and explain the operation of IDQ QRNG. Then, we present two limitations of IDQ QRNG and describe the differences between IDQ QRNG and our entropy accumulation. The second comparison is with Fast-Refresh of Windows RNG. We compare the iteration number *l*, which is obtained when applying each entropy accumulation. The iteration number of Windows RNG is calculated using the theory in [2]. Note that without some additional components, the theory in [2] cannot be directly applied. Section 5 is the conclusion.

## 2. Theoretical Background and Main Theorem

In this section, we describe the theoretical background of entropy accumulation using only the XOR operation. In particular, we use the following notation:$Z_{m}^{n}$: Direct product of *n* copies of the group $Z_{m}$. Note that the bitwise XOR operation corresponds to the + operation over $Z_{2}^{n}$.$F(Z_{m}^{n})$: the space of all complex valued functions on $Z_{m}^{n}$.$\Gamma^{\left(l\right)}=\sum_{j=1}^{l}Y_{j}$, where $Y_{j}\in Z_{2}^{n}$ is the *n*-bit random variable that represents the input sequence.${∥f∥}_{min}=\min\left\{\left|f\left(x\right)\right|:x\in Z_{m}^{n}\right\}$, $f\in F(Z_{m}^{n})$.${∥f∥}_{\infty }=\max\left\{\left|f\left(x\right)\right|:x\in Z_{m}^{n}\right\}$, $f\in F(Z_{m}^{n})$.${∥f∥}_{1}=\sum_{x\in Z_{m}^{n}}\left|f\left(x\right)\right|,f\in F\left(Z_{m}^{n}\right).$$D_{X}$: probability distribution of the random variable X.$H_{min}\left(D_{X}\right)=-\log_{2}{∥D_{X}∥}_{\infty }$. $H_{min}\left(D_{X}\right)$ implies the min-entropy of $D_{X}$.

We show that as the number of input sequences required to generate one output sequence, represented by *l*, approaches infinity, $H_{min}\left(D_{\Gamma^{\left(l\right)}}\right)$ converges to *n*. Furthermore, we provide the optimal value of *l* necessary to surpass the specified min-entropy α(<n). First, we provide a solution for the case n=1 and explain why this solution is inappropriate for the general *n*-bit case. Thereafter, we provide a general solution using a Discrete Fourier Transform and Convolution.

First, we show why the problem we are trying to solve is challenging. The difficult point of our problem is that in order to determine the value of $D_{\Gamma^{\left(l\right)}}$, complex linear operations must be performed on the function values of $D_{Y_{j}}$. For example, suppose n=2,   $Y_{1},Y_{2},Y_{3}$ are independent and $D_{Y_{1}},D_{Y_{2}},D_{Y_{3}}$ are identical to the distribution *D*. The distribution *D* is determined as $D\left(0,0\right)=\frac{1}{8},\;D\left(0,1\right)=\frac{1}{4},\;D\left(1,0\right)=\frac{3}{8},\;D\left(1,1\right)=\frac{1}{4}$. Let us calculate $D_{\Gamma^{\left(3\right)}}$, which suffices to show the complexity of computation.
$$\begin{array}{c} D_{\Gamma^{\left(3\right)}}\left(0,0\right)=\sum_{x⊕y⊕z=(0,0)}D_{Y_{1}}\left(x\right)D_{Y_{2}}\left(y\right)D_{Y_{3}}\left(z\right)=D\left(0,0\right)D\left(0,0\right)D\left(0,0\right)+\\ D\left(0,0\right)D\left(0,1\right)D\left(0,1\right)+\cdots +D\left(1,1\right)D\left(1,1\right)D\left(0,0\right)=\frac{124}{512}\approx 0.242.\\ D_{\Gamma^{\left(3\right)}}\left(0,1\right)=\sum_{x⊕y⊕z=(0,1)}D_{Y_{1}}\left(x\right)D_{Y_{2}}\left(y\right)D_{Y_{3}}\left(z\right)=D\left(0,0\right)D\left(0,0\right)D\left(0,1\right)+\\ D\left(0,0\right)D\left(0,1\right)D\left(0,0\right)+\cdots +D\left(1,1\right)D\left(1,1\right)D\left(0,1\right)=\frac{128}{512}=0.25. \end{array}$$
$$\begin{array}{c} D_{\Gamma^{\left(3\right)}}\left(1,0\right)=\sum_{x⊕y⊕z=(1,0)}D_{Y_{1}}\left(x\right)D_{Y_{2}}\left(y\right)D_{Y_{3}}\left(z\right)=D\left(0,0\right)D\left(0,0\right)D\left(1,0\right)+\\ D\left(0,0\right)D\left(0,1\right)D\left(1,1\right)+\cdots +D\left(1,1\right)D\left(1,1\right)D\left(1,0\right)=\frac{132}{512}\approx 0.258.\\ D_{\Gamma^{\left(3\right)}}\left(1,1\right)=\sum_{x⊕y⊕z=(1,1)}D_{Y_{1}}\left(x\right)D_{Y_{2}}\left(y\right)D_{Y_{3}}\left(z\right)=D\left(0,0\right)D\left(0,0\right)D\left(1,1\right)+\\ D\left(0,0\right)D\left(0,1\right)D\left(1,0\right)+\cdots +D\left(1,1\right)D\left(1,1\right)D\left(1,1\right)=\frac{128}{512}=0.25. \end{array}$$

From the above calculations, we derive two features. First, to calculate one function value of $D_{\Gamma^{\left(l\right)}}$, we must sum $2^{(l-1)n}$ terms. That is, to calculate
(1)$$D_{\Gamma^{\left(l\right)}}\left(x\right)=\sum_{x_{1}⊕x_{2}⊕\cdots ⊕x_{l}=x}D_{Y_{1}}\left(x_{1}\right)D_{Y_{2}}\left(x_{2}\right)\cdots D_{Y_{l}}\left(x_{l}\right),$$
the first l−1 terms $x_{1},x_{2},\cdots x_{l-1}$ could be any value and the last term $x_{l}$ is automatically determined by equation $x_{1}⊕x_{2}⊕\cdots ⊕x_{l}=x$. Because there is $2^{n}$ choices respectively, the total terms would be $2^{(l-1)n}$; however, it is difficult to calculate. Second, as *l* grows, $D_{\Gamma^{\left(l\right)}}$ tends to uniform distribution. The distance between the original distribution *D* and the uniform distribution *I* with respect to infinite norm ${∥D-I∥}_{\infty }$ of above example is $\frac{1}{8}$. However, we can observe that ${∥D_{\Gamma^{\left(3\right)}}-I∥}_{\infty }$ is $\frac{1}{128}$. As *l* grows, the terms that should be computed to calculate the function value grow rapidly; consequently, the impact of one function value will decrease. Although this phenomenon seems natural, still, the following questions remain: Under what conditions does this convergence happen? How about the convergence rate? How can we prove the related results?

### 2.1. Entropy Accumulation with n=1

Let us consider the relatively simple case of n=1 and $Y_{j}$ following an independent and identical distribution (IID). In this case, all $Y_{j}$ follow the same distribution $D(=D_{Y_{j}})$ and a recursive relationship $\Gamma^{\left(j+1\right)}=\Gamma^{\left(j\right)}⊕Y_{j+1}$ is established. Thus, we can express $D_{\Gamma^{\left(j+1\right)}}\left(1\right)$ based on the following relationship: (2)$$D_{\Gamma^{\left(j+1\right)}}\left(1\right)=D_{\Gamma^{\left(j\right)}}\left(1\right)\left[1-D_{Y_{j+1}}\left(1\right)\right]+\left[1-D_{\Gamma^{\left(j\right)}}\left(1\right)\right]D_{Y_{j+1}}\left(1\right).$$(2) is derived based on the property that the sum of two bits resulting in 1 can be obtained by adding 1 and 0, or 0 and 1. Moreover, $D_{\Gamma^{\left(j+1\right)}}\left(1\right)$ in (2) can be interpreted as a point where $D_{\Gamma^{\left(j\right)}}\left(1\right)$ and $1-D_{\Gamma^{\left(j\right)}}\left(1\right)$ are internalized into $1-D_{Y_{j+1}}\left(1\right):D_{Y_{j+1}}\left(1\right)$. Because $D_{\Gamma^{\left(j\right)}}\left(1\right)$ and $1-D_{\Gamma^{\left(j\right)}}\left(1\right)$ are symmetric about $x=\frac{1}{2}$, the condition 0<D(1)<1 causes the convergence of $D_{\Gamma^{\left(j\right)}}\left(1\right)$ to $\frac{1}{2}$ (i.e., the maximum possible entropy) as *j* increases. Figure 5 illustrates the scenario.

A more specific formula exists to accurately illustrate this situation. The following lemma, often referred to as the “Piling Up Lemma,” further details this [16].

**Fact** **1**(Piling Up Lemma [16])**.** *Let $Y_{1},Y_{2},\cdots ,Y_{l}$ be independent one-bit random variables, and let $\Gamma^{\left(l\right)}:=\sum_{j=1}^{l}Y_{j}$. Then,*
$$D_{\Gamma^{\left(l\right)}}\left(0\right)=\frac{1}{2}+2^{l-1}\prod_{j=1}^{l}\left[D_{Y_{j}}\left(0\right)-\frac{1}{2}\right].$$

Because $0<\left|D_{Y_{j}}\left(0\right)-\frac{1}{2}\right|<\frac{1}{2}$ for each *j*, $2^{l-1}\prod_{j=1}^{l}\left[D_{Y_{j}}\left(0\right)-\frac{1}{2}\right]$ converges to 0 and $D_{\Gamma^{\left(l\right)}}\left(0\right)$ converges to 1/2 as *l* approaches infinity.

This equation cannot be applied when *n* is greater than 2. When n=2 or more, the probability that each bit produces 0 or 1 converges to 1/2; however, we cannot sum up the min-entropies of each position to calculate the total min-entropy because it is allowed only when all bits are independent of each other. Therefore, a new method is required for addressing these problems.

### 2.2. Convolution and Discrete Fourier Transform

In this subsection, we describe techniques applicable in the special case where *n* equals 1, as well as in more general cases. First, we reformulated the problem using the concept of convolution.

**Definition** **1**(Convolution)**.** *The Convolution of $f,g\in F(Z_{m}^{n})$ is defined as*
$$f*g\left(x\right):=\sum_{y\in Z_{m}^{n}}f\left(x-y\right)g\left(y\right).$$

For the entropy accumulation problem of interest, m=2. Using the language of convolution, (1) becomes $D_{\Gamma^{\left(l\right)}}=D_{Y_{1}}*D_{Y_{2}}*\cdots *D_{Y_{l}}$. The entropy accumulation problem is reduced to a problem of handling this convolution. Fortunately, there exists a mathematical concept, the “Fourier Transform”, that harmonizes well with convolution.

**Definition** **2**(Discrete Fourier Transform)**.** *The Discrete Fourier Transform of $f\in F(Z_{m}^{n})$ is defined as*
$$\hat{f}\left(t\right):=\sum_{x\in Z_{m}^{n}}f\left(x\right)e^{-\frac{2\pi i}{m}x\cdot t}.$$

A Discrete Fourier Transform is the mapping from $F(Z_{m}^{n})$ to $F(Z_{m}^{n})$. In fact, this transform is one-to-one mapping. Proposition 1 supports this.

**Lemma** **1.**
*If t≠0, $\sum_{x\in Z_{m}^{n}}e^{-\frac{2\pi i}{m}x\cdot t}=0$.*


**Proof.** First, note that
$$\begin{array}{c} \sum_{x\in Z_{m}^{n}}e^{-\frac{2\pi i}{m}x\cdot t}=\sum_{x\in Z_{m}^{n}}e^{-\frac{2\pi i}{m}x\cdot t\left(\mathrm{mod}m\right)}. \end{array}$$We define $\phi_{\;t}:Z_{m}^{n}\to Z_{m}$ as:
$$\begin{array}{c} \phi_{\;t}\left(x\right):=x\cdot t\left(\mathrm{mod}m\right). \end{array}$$Then, $\phi_{\;t}$ is a homomorphism:
$$\begin{array}{c} \phi_{\;t}\left(x+y\right)=\left(x+y\right)\cdot t\left(\mathrm{mod}m\right)=x\cdot t\left(\mathrm{mod}m\right)+y\cdot t\left(\mathrm{mod}m\right)=\phi_{\;t}\left(x\right)+\phi_{\;t}\left(y\right). \end{array}$$As t≠0, $\;\phi_{\;t}^{-1}\left(0\right)\neq Z_{m}^{n}$. Therefore, for every $s\in Z_{m}$, $\;\phi_{\;t}^{-1}\left(s\right)$ contains the same number of elements. Let the number of elements be *N*, then
$$\begin{array}{c} \sum_{x\in Z_{m}^{n}}e^{-\frac{2\pi i}{m}x\cdot t}=\sum_{x\in Z_{m}^{n}}e^{-\frac{2\pi i}{m}\phi_{\;t}\left(x\right)}=N\sum_{s\in Z_{m}}e^{-\frac{2\pi i}{m}s}=0. \end{array}$$The last equality holds because each $e^{-\frac{2\pi i}{m}s}$ is the root of the complex equation $z^{m}-1=0$.    □

**Lemma** **2.**
*Let f be the element in $F(Z_{m}^{n})$ and $\hat{f}$ be the Fourier Transform function. Then,*

$$\frac{1}{m^{n}}\sum_{t\in Z_{m}^{n}}\hat{f}\left(t\right)e^{\frac{2\pi i}{m}x\cdot t}=f\left(x\right).$$



**Proof.** By Lemma 1, $\;\sum_{t\in Z_{m}^{n}}f\left(s\right)e^{-\frac{2\pi i}{m}\left(x-s\right)\cdot t}=0$ if s≠x. Therefore,
$$\begin{array}{c} \frac{1}{m^{n}}\sum_{s\in Z_{m}^{n}}\sum_{t\in Z_{m}^{n}}f\left(s\right)e^{-\frac{2\pi i}{m}\left(x-s\right)\cdot t}=\frac{1}{m^{n}}\sum_{t\in Z_{m}^{n}}f\left(x\right)e^{-\frac{2\pi i}{m}\left(x-x\right)\cdot t}=\frac{1}{m^{n}}\sum_{t\in Z_{m}^{n}}f\left(x\right)=f\left(x\right). \end{array}$$   □

**Proposition** **1.**
*Let f and g be the elements of $F(Z_{m}^{n})$. If $\hat{f}=\hat{g}$, f=g.*


**Proof.** We assume that $\hat{f}=\hat{g}$. Then,
$$\begin{array}{c} f\left(x\right)=\frac{1}{m^{n}}\sum_{t\in Z_{m}^{n}}\hat{f}\left(t\right)e^{\frac{2\pi i}{m}x\cdot t}=\frac{1}{m^{n}}\sum_{t\in Z_{m}^{n}}\hat{g}\left(t\right)e^{\frac{2\pi i}{m}x\cdot t}=g\left(x\right) \end{array}$$
holds for every $x\in Z_{m}^{n}$ from Lemma 2.   □

The following theorem asserts that the convolution product of functions is represented as a multiplication in the transformed space. This plays a significant role in proving our main theorem.

**Proposition** **2.**
*Let f and g be the elements of $F(Z_{m}^{n})$. Then, $\hat{f*g}=\hat{f}\hat{g}$.*


**Proof.** By the definitions of convolution and Discrete Fourier Transform,
$$\begin{array}{cc} \hat{f*g}\left(t\right)\; & =\sum_{x\in Z_{m}^{n}}\sum_{y\in Z_{m}^{n}}f\left(x-y\right)g\left(y\right)e^{-\frac{2\pi i}{m}x\cdot t}\\ \; & =\sum_{x\in Z_{m}^{n}}\sum_{y\in Z_{m}^{n}}f\left(x-y\right)e^{-\frac{2\pi i}{m}\left(x-y\right)\cdot t}g\left(y\right)e^{-\frac{2\pi i}{m}y\cdot t}\\ \; & =\sum_{x\in Z_{m}^{n}}f\left(x-y\right)e^{-\frac{2\pi i}{m}\left(x-y\right)\cdot t}\sum_{y\in Z_{m}^{n}}g\left(y\right)e^{-\frac{2\pi i}{m}y\cdot t}\\ \; & =\sum_{x\in Z_{m}^{n}}f\left(x\right)e^{-\frac{2\pi i}{m}x\cdot t}\sum_{y\in Z_{m}^{n}}g\left(y\right)e^{-\frac{2\pi i}{m}y\cdot t}\\ \; & =\hat{f}\left(t\right)\hat{g}\left(t\right). \end{array}$$   □

To intuitively determine why $\Gamma^{\left(l\right)}$ converges to a uniform distribution, we must understand both the properties of the Discrete Fourier Transform applied to the distribution and the Discrete Fourier Transform of a uniform distribution.

**Proposition** **3.**
*Let D be an arbitrary probability distribution and I be a uniform distribution of $Z_{m}^{n}$. Then,*
*(i)*      
*For all $t\in Z_{m}^{n},\left|\hat{D}\left(t\right)\right|\leq 1$.*
*(ii)*     
*$\hat{D}\left(0\right)=1$.*
*(iii)*    
*For all $t\in Z_{m}^{n},\hat{I}\left(t\right)=\delta_{\;t,0}$.*


*The symbols $\delta_{\;t,0}$ denote the Kronecker delta. The Kronecker delta is defined as 1 when ***t*** is zero vector and 0 for all other ***t***.*


**Proof.** Proof of (a):
$$\begin{array}{cc} \; & \left|\hat{D}\left(t\right)\right|=\left|\sum_{x\in Z_{m}^{n}}D\left(x\right)e^{-\frac{2\pi i}{m}x\cdot t}\right|\leq \sum_{x\in Z_{m}^{n}}\left|D\left(x\right)e^{-\frac{2\pi i}{m}x\cdot t}\right|=\sum_{x\in Z_{m}^{n}}D\left(x\right)=1. \end{array}$$Proof of (b):
$$\begin{array}{cc} \; & \hat{D}\left(0\right)=\sum_{x\in Z_{m}^{n}}D\left(x\right)e^{-\frac{2\pi i}{m}x\cdot 0}=\sum_{x\in Z_{m}^{n}}D\left(x\right)=1. \end{array}$$Proof of (c): We know from part (a) that $\hat{I}\left(0\right)=1$. For the remaining t≠0,
$$\begin{array}{cc} \; & \hat{I}\left(t\right)=\sum_{x\in Z_{m}^{n}}I\left(x\right)e^{-\frac{2\pi i}{m}x\cdot t}=\frac{1}{m^{n}}\sum_{x\in Z_{m}^{n}}e^{-\frac{2\pi i}{m}x\cdot t}=0. \end{array}$$The last equality is based on Lemma 1.    □

From Proposition 2, we have $\hat{D_{\Gamma^{\left(l\right)}}}=\hat{D_{Y_{1}}}\hat{D_{Y_{2}}}\cdots \hat{D_{Y_{l}}}$. By Proposition 3, $\hat{D_{\Gamma^{\left(l\right)}}}\left(0\right)=1$, whereas for t≠0, the value of $\hat{D_{\Gamma^{\left(l\right)}}}\left(t\right)$ approaches 0 as *l* increases. Specifically, $\hat{D_{\Gamma^{\left(l\right)}}}$ converges to $\delta_{,t,0}$ as *l* increases. Since the Discrete Fourier Transform is a one-to-one function by Proposition 1, we can infer that $D_{\Gamma^{\left(l\right)}}$ converges to *I* as *l* increases.

### 2.3. Main Theorem

In the previous subsection, we confirmed that $D_{\Gamma^{\left(l\right)}}$ converges to a uniform distribution *I* as *l* increases. In this subsection, we present a solution to the entropy accumulation problem based on this approach. Specifically, we aim to find a condition for the random variable $Y_{j}$ and a value for *l* such that $D_{\Gamma^{\left(l\right)}}$ achieves a specific min-entropy. The following theorem is one of the main results of our study:

**Theorem** **1.**
*Let $Y_{1},Y_{2},\cdots Y_{l}$ be independent n-bit random variables, and $\Gamma^{\left(l\right)}:=Y_{1}⊕Y_{2}⊕\cdots ⊕Y_{l}$. We define $\omega :=\min\{{∥D_{Y_{1}}∥}_{min},{∥D_{Y_{2}}∥}_{min},\cdots ,{∥D_{Y_{l}}∥}_{min}\}$. Then,*

$$H_{min}\left(D_{\Gamma^{\left(l\right)}}\right)\geq n-\log_{2}\left[1+\left(2^{n}-1\right){\left(1-2^{n}\omega \right)}^{l}\right]\approx n-\frac{1}{\ln2}\left(2^{n}-1\right){\left(1-2^{n}\omega \right)}^{l}.$$



Note that the condition for *n*-bit random variables $Y_{1},Y_{2},\cdots Y_{l}$ is not an IID. Because the above theorem only requires the independence of random variables without the condition of identical distribution, it can be effectively applied when using parallel entropy sources. We provide the proof of Theorem 1.

**Proof.** For any function $f\in L(Z_{2}^{n})$, we have
(3)$$\hat{f}\left(t\right)=\sum_{x\in Z_{2}^{n}}f\left(x\right)e^{\pi i(x\cdot t)}=\sum_{x\in Z_{2}^{n}}f\left(x\right){\left(-1\right)}^{x\cdot t},$$
(4)$$\frac{1}{2^{n}}\sum_{t\in Z_{m}^{n}}\hat{f}\left(t\right)e^{\pi i(x\cdot t)}=\frac{1}{2^{n}}\sum_{t\in Z_{m}^{n}}\hat{f}\left(t\right){\left(-1\right)}^{x\cdot t}=f\left(x\right).$$This is obtained from Lemma 2 with m=2. Using the function $\phi_{\;t}$ of Lemma 2, (3) and (4) can be written as
(5)$$\hat{f}\left(t\right)=\sum_{x\in Z_{2}^{n}}f\left(x\right)e^{\pi i(x\cdot t)}=\sum_{x\in Z_{2}^{n}}f\left(x\right){\left(-1\right)}^{x\cdot t}=\sum_{x\in \phi_{\;t}^{-1}\left(0\right)}f\left(x\right)-\sum_{x\in \phi_{\;t}^{-1}\left(1\right)}f\left(x\right),$$

(6)
$$\frac{1}{2^{n}}\sum_{t\in Z_{2}^{n}}\hat{f}\left(t\right)e^{\pi i(x\cdot t)}=\frac{1}{2^{n}}\sum_{t\in Z_{2}^{n}}\hat{f}\left(t\right){\left(-1\right)}^{x\cdot t}=\frac{1}{2^{n}}\left[\sum_{t\in \phi_{\;x}^{-1}\left(0\right)}\hat{f}\left(t\right)-\sum_{t\in \phi_{\;x}^{-1}\left(1\right)}\hat{f}\left(t\right)\right]=f\left(x\right).$$

We apply (5) to each $D_{Y_{j}}$ with t≠0. Because $\sum_{x\in Z_{2}^{n}}D_{Y_{j}}\left(x\right)=1$ and $D_{Y_{j}}\left(x\right)>\omega >0$,
(7)$$\begin{array}{cc} \left|\hat{D_{Y_{j}}}\left(t\right)\right| & =\left|\sum_{x\in \phi_{\;t}^{-1}\left(0\right)}D_{Y_{j}}\left(x\right)-\sum_{x\in \phi_{\;t}^{-1}\left(1\right)}D_{Y_{j}}\left(x\right)\right|\\ \; & =\left|\sum_{x\in \phi_{\;t}^{-1}\left(0\right)}\left[D_{Y_{j}}\left(x\right)-\omega \right]-\sum_{x\in \phi_{\;t}^{-1}\left(1\right)}\left[D_{Y_{j}}\left(x\right)-\omega \right]\right|\\ \; & \leq \left|\sum_{x\in Z_{2}^{n}}\left[D_{Y_{j}}\left(x\right)-\omega \right]\right|=1-2^{n}\omega . \end{array}$$From Theorems (2) and (7),
$$\begin{array}{cc} \left|D_{\Gamma^{\left(l\right)}}\left(t\right)\right|\; & =\frac{1}{2^{n}}\left|\sum_{t\in \phi_{\;x}^{-1}\left(0\right)}\hat{D_{\Gamma^{\left(l\right)}}}\left(t\right)-\sum_{t\in \phi_{\;x}^{-1}\left(1\right)}\hat{D_{\Gamma^{\left(l\right)}}}\left(t\right)\right|\\ \; & \leq \frac{1}{2^{n}}\sum_{t\in Z_{2}^{n}}\left|\hat{D_{\Gamma^{\left(l\right)}}}\left(t\right)\right|=\frac{1}{2^{n}}\sum_{t\in Z_{2}^{n}}\prod_{j=1}^{l}\left|\hat{D_{Y_{j}}}\left(t\right)\right|\\ \; & =\frac{1}{2^{n}}\left[\prod_{j=1}^{l}\left|\hat{D_{Y_{j}}}\left(0\right)\right|+\sum_{t\in Z_{2}^{n}/\left\{0\right\}}\prod_{j=1}^{l}\left|\hat{D_{Y_{j}}}\left(t\right)\right|\right]\\ \; & \leq \frac{1}{2^{n}}\left[1+\left(2^{n}-1\right){\left(1-2^{n}\omega \right)}^{l}\right]. \end{array}$$Therefore,
$$\begin{array}{cc} H_{min}\left(D_{\Gamma^{\left(l\right)}}\right)\; & =\max\{-\log_{2}\left|D_{\Gamma^{\left(l\right)}}\left(t\right)\right|:t\in Z_{2}^{n}\}\\ \; & \geq n-\log_{2}\left[1+\left(2^{n}-1\right){\left(1-2^{n}\omega \right)}^{l}\right]\\ \; & \approx n-\frac{1}{\ln2}\left(2^{n}-1\right){\left(1-2^{n}\omega \right)}^{l}. \end{array}$$The final approximation is based on the Taylor theorem.    □

## 3. Applying Theorem 1 to Image Sensor-Based RNG

In this section, we describe the process of applying Theorem 1 to an image sensor-based random number generator. First, we describe the process of generating the input sequences from the entropy sources of the image sensor. Subsequently, we verify whether the generated input sequences satisfy the assumptions of Theorem 1. Next, we establish the lower bound for the min-entropy, which is considered secure based on three standards. Then, we provide the theoretical number of iterations required to achieve a min-entropy higher than the established lower bound. Furthermore, we validate our theory using experimental results. Finally, we estimate the entropy accumulation speed based on frames per second (FPS) of the image sensors used. Thereafter, we compare and analyze the random number generation speed of our system with that of the ID Quantique’s QRNG chip.

### 3.1. Image Sensor-Based RNG

We use the ‘PV 4209K’ image sensor, which utilizes 11,520 optical black pixels (OBP) as physical entropy sources. Each OBP of the image sensor transmits 2-bit data to a PC. The PC sequentially stores the 2-bit data transmitted by the multiple OBPs. See Figure 6.

### 3.2. Experimentation Process for Entropy Accumulation

Before describing the entropy accumulation experiments, we use the following notation:$W_{i}$: A random variable corresponding to the 2-bit data of the *i*-th optical black pixel (OBP). “If the value of *i* reaches the last pixel (11,520), the next value of *i* refers to the first pixel.”$Y_{j}:=W_{4j-3}∥W_{4j-2}∥W_{4j-1}∥W_{4j}.$ For example, if $W_{1}=\left(0,1\right)$, $W_{2}=\left(1,1\right)$, $W_{3}=\left(0,0\right)$, $W_{4}=\left(1,0\right)$, $Y_{1}$ becomes $Y_{1}=\left(0,1,1,1,0,0,1,0\right)$.$\Gamma_{k}^{\left(l\right)}:=\sum_{j=(kl-l+1)}^{kl}Y_{j}$. This refers to the *k*-th output sequence, which is generated by adding (XOR) *l* input sequences.

To experimentally validate entropy accumulation, we utilized the verification method outlined in [17]. Ref. [17] is a min-entropy estimation tool, which estimates the min-entropy of output sequences by collecting 1,000,000 *n*-bit output sequences. However, there is a requirement that the value of *n* must be at least 8. Therefore, to satisfy this condition, we created new 8-bit data $Y_{j}$ by concatenating four 2-bit datasets $W_{4j-3}$, $W_{4j-2}$, $W_{4j-1}$, $W_{4j}$, and $Y_{j}$ becomes an input sequence. This process is shown in Figure 7.

After setting $Y_{j}$, we select the XOR operation iteration number (l−1) and accumulate $\Gamma_{k}^{\left(l\right)}$ in the entropy pool. This process is illustrated in Figure 8.

The number *l* is determined using Theorem 1. After collecting $\Gamma_{k}^{\left(l\right)}$(1≤k≤ 1,000,000) in the entropy pool, we use [17] to verify the min-entropy.

### 3.3. Setting Lower Bound of Min-Entropy

We provide three evaluation criteria that can be used to determine the lower bound of the min-entropy, which a true random number generator must exceed, acquired through the entropy accumulation process.

#### 3.3.1. Maximum Value of the Most Common Value Estimate

In [17], when the output sequences are determined to follow the IID, the Most Common Value Estimate is assumed to be the min-entropy of the output sequences [17]. However, there is an upper bound on the min-entropy value in the Most Common Value Estimate. Regardless of the output sequences used for the test, this upper bound cannot be exceeded. The Most Common Value Estimate estimates the min-entropy as described in Algorithm 1.
**Algorithm 1** Most Common Value Estimate**Input:** $S=(s_{1},\ldots ,s_{L})$, *L*: length of *S*, $S_{i}\in {\left\{0,1\right\}}^{n}$(1≤i≤L)**Output:** min-entropy of dataset $S:H_{min}$      1:Calculate the mode of *S*. We denote this value by MODE      2:$\hat{p}=\frac{MODE}{L}$      3:$p_{u}=\min\left(1,\hat{p}+2.576\sqrt{\frac{\hat{p}\left(1-\hat{p}\right)}{L-1}}\right)$      4:$H_{min}=-\log_{2}p_{u}$

To compute the upper bound of the Most Common Value Estimate for an 8-bit dataset *S*, where the length of *S* is 1,000,000, the mode of *S* should be one. That is, $\hat{p}=1/256$. Using $\hat{p}$, we obtain $H_{min}=7.94$. Therefore, it is reasonable for the lower bound of the min-entropy to not exceed 7.94.

#### 3.3.2. True Random 8-Bit in [17]

Ref. [17] provides True Random Data in 1-bit, 4-bit, and 8-bit units as samples for evaluating min-entropy. From Figure 9, it can be confirmed that the min-entropy of true random 8-bit data in [17] is approximately 7.86.

#### 3.3.3. Criterion of Min-Entropy by BSI AIS 20/31 [18]

The Federal Office for Information Security in Germany (Bundesamt für Sicherheit in der Informationstechni, BSI) asserts in the AIS 20/31 document that the output sequences of a cryptographic random number generator should have a min-entropy of 0.98 per bit. In the case of 8-bit data, 7.84 becomes the lower bound of entropy.

From the three criteria mentioned above, we have chosen 7.86 as the min-entropy lower bound. This value, which is smaller than 7.94 and more stringent than 7.84, appears to be a valid choice for the lower bound.

### 3.4. Applying Theorem 1 to Input Sequences

In this subsection, we use Theorem 1 and obtain *l*. Instead of directly applying Theorem 1 to $Y_{j}$, we employ a “divide and conquer” approach to compute the total min-entropy. Before explaining this strategy, note that each $W_{i}$ can be regarded as an independent random variable. This assumption is reasonable because all pixels can be considered independent entropy sources.

Because we conducted the experiment with eight bits, *n* must be eight when applying Theorem 1. However, this results in the following problem: the required number *l* is exceedingly large. To address this issue, we exploit the fact that $Y_{j}$ is constructed by concatenating four independent entropy sources. $\Gamma_{k}^{\left(l\right)}$ is generated by considering the XOR of *l* instances in $Y_{j}$. Therefore, if we break down $\Gamma_{k}^{\left(l\right)}$ into two bits, then $\Gamma_{k}^{\left(l\right)}$ can be considered as four concatenations of the XOR of *l* instances of $W_{i}$. This can be expressed by the following formula:$$\begin{array}{c} \Gamma_{k}^{\left(l\right)}=\sum_{j=(kl-l+1)}^{kl}W_{4j-3}∥W_{4j-2}∥W_{4j-1}∥W_{4j}=\\ \left(\sum_{j=(kl-l+1)}^{kl}W_{4j-3}\right)∥\left(\sum_{j=(kl-l+1)}^{kl}W_{4j-2}\right)∥\left(\sum_{j=(kl-l+1)}^{kl}W_{4j-1}\right)∥\left(\sum_{j=(kl-l+1)}^{kl}W_{4j}\right). \end{array}$$

The four parts of $\Gamma_{k}^{\left(l\right)}$ are independent of each other. Therefore, we calculated the total min-entropy by individually determining the min-entropy for each of the four parts and then summing them up. If the min-entropy of each 2-bit segment of $\Gamma_{k}^{\left(l\right)}$ is greater than 1.965, $H_{min}\left(\Gamma_{k}^{\left(l\right)}\right)$ will be greater than 7.86. Theorem 1 is applied to the four 2-bit segments of $\Gamma_{k}^{\left(l\right)}$. To apply Theorem 1, the following two conditions must be satisfied. The first pertains to the independence of $W_{i}$. The second condition states that the value of ω should exceed zero. For the first condition, we established that each $W_{i}$ could be regarded as an independent random variable. For the second condition, we can estimate the value of ω by analyzing the probability distribution of the data transmitted by each OBP. We constructed the probability distribution of each $W_{i}$ using 2-bit data transmitted by each of the 11,520 OBPs over 2000 transmissions. From the obtained distribution, we can confirm that the value of ${∥D_{W_{i}}∥}_{min}$ is greater than 0.075 for all *i*. Figure 10 illustrates the probability distribution of four randomly selected OBPs. It can be observed that each number has appeared at least 150 times.

Therefore, we estimate that ω is at least 0.075. From Theorem 1, the inequality
$$\begin{array}{c} 2-\frac{1}{\ln2}\left(2^{2}-1\right){\left(1-2^{2}\omega \right)}^{l}\geq 1.965 \end{array}$$
provides the number *l* necessary for the min-entropy of each 2-bit segment of $\Gamma_{k}^{\left(l\right)}$ to exceed 1.965.
$$\begin{array}{cc} \; & 2-\frac{1}{\ln2}\left(2^{2}-1\right){\left(1-2^{2}\omega \right)}^{l}\geq \;1.965\\ \; & \Rightarrow \frac{0.035\cdot \ln2}{3}\geq {\left(0.7\right)}^{l}\\ \; & \Rightarrow \frac{\ln\left(\frac{0.035\cdot \ln2}{3}\right)}{\ln(0.7)}\leq \;l\\ \; & \Rightarrow 15.845\leq l. \end{array}$$

From the last inequality, we can conclude that if we use 15 XOR operations to create $\Gamma_{k}^{\left(l\right)}$, $H_{min}\left(\Gamma_{k}^{\left(l\right)}\right)$ exceeds 7.86.

### 3.5. Experimental Result

We describe the experimental validation of the results in Section 3.4. We predicted that through 15 XOR operations for entropy accumulation, more than 7.86-bit of entropy per 8-bit would be guaranteed. Upon conducting actual experiments, it was confirmed that even with only four XOR operations, more than 7.86 min-entropy per 8-bit was accomplished. Table 1 presents the experimental results.

The experimental results are analyzed as follows. When l=1, it refers to the case where the XOR operation is not used, and the min-entropy per 8-bit is 3.305, which is lower than the min-entropy calculated based on the probability distribution of the 2-bit from the pixels.

Min-entropy depends on the maximum value of the probability distribution function. By observing the 2-bit probability distribution, we confirmed that the maximum value of the probability distribution function was 0.425. This can also be confirmed from Figure 10, where all the numbers (0, 1, 2, 3) in the four randomly selected distributions do not exceed 850. Therefore, when calculating the min-entropy of 2-bit, the lower bound of $H_{min}\left(D_{W_{i}}\right)$ is 1.2344, and the lower bound of $H_{min}\left(D_{Y_{j}}\right)$ created by concatenating the four $D_{W_{i}}$ is 4.9378.

The min-entropy estimated through the probability distribution is lower than that estimated by [17] because of the conservative measurement method of [17]. In [17], it is determined whether the output sequence follows the IID track or the non-IID track before measuring min-entropy. If the output sequence follows the IID track, the min-entropy measured by the most common value estimation method becomes the final min-entropy of the output sequence. However, if the output sequence follows the non-IID track, the smallest value among the min-entropies measured by the other 10 methods, including the Most Common Value Estimate, becomes the final min-entropy of the output sequence. For l=1, it was confirmed that the original output sequence followed a non-IID track. Therefore, the min-entropy value estimated by [17] was smaller than that calculated based on the probability distribution.

As the value of *l* increases, it can be observed that the increment of min-entropy decreases. This is because the maximum possible min-entropy value is 7.94, and as it approaches this number, the amount of min-entropy that can be increased by increasing the number of XOR operations becomes small. The greatest change in the min-entropy value occurs when *l* changes from l=1 to l=2. This is because the output sequence follows the IID track if l≥2.

## 4. Comparing with Other Entropy Accumulations

In this section, we compare and analyze the entropy accumulation we developed with other entropy accumulations. The first comparison is with the Slow-Refresh used in IDQ QRNG. The second comparison target is the Fast-Refresh of Windows RNG. In comparing with the Slow-Refresh, we focus on the differences in the accumulation mechanism. In comparing with the Fast-Refresh, the analysis is carried out by calculating the iteration number of operations of Fast-Refresh in our experimental environment.

### 4.1. Comparing with the Slow-Refresh of IDQ QRNG

IDQ QRNG uses matrix multiplication as the entropy accumulation function [19,20]. For the analysis of this method, we first explain the theoretical background of entropy accumulation and the “Leftover Hash Lemma” and then explain the entropy accumulation of IDQ QRNG. We also check whether the entropy accumulation of IDQ QRNG meets the conditions of the Leftover Hash Lemma. Lastly, we summarize the differences between our entropy accumulation and that of IDQ QRNG.

#### 4.1.1. Leftover Hash Lemma [3,19]

The Leftover Hash Lemma is a theorem that ensures the conversion of a low min-entropy input sequence into a high min-entropy output sequence. In order to state the Leftover Hash Lemma, we need two key concepts: the 2-universal hash family and statistical distance.

**Definition** **3**(Statistical distance)**.** *Let X and $X^{\prime }$ be random variables that take values in same set. The statistical distance between two probability distributions $D_{X}$ and $D_{X^{\prime }}$ is defined as*
$$\Delta_{D_{X},D_{X^{\prime }}}=\frac{1}{2}{∥D_{X}-D_{X^{\prime }}∥}_{1}.$$

**Definition** **4**(2-universal hash family)**.** *Let Y be a random variable uniformly distributed over S. A family ${\left\{f_{s}:T\to V\right\}}_{s\in S}$ of hash functions is called 2-universal if for any distinct inputs $x\neq x^{\prime }$*
$$Pr\left[f_{Y}\left(x\right)=f_{Y}\left(x^{\prime }\right)\right]\leq \frac{1}{\left|V\right|}.$$

Based on these definitions, we can state the Leftover Hash Lemma.

**Fact** **2**(Leftover Hash Lemma)**.** *Let ${\left\{f_{s}:T\to V\right\}}_{s\in S}$ be the 2-universal hash family. Let X and Y be independent random variables, where Y is uniformly distributed over S, and X takes values in T. Let $U_{S\times V}$ be the uniform distribution on S × V. Then,*
$$\Delta_{D_{\left(Y,f_{Y}\left(X\right)\right)},U_{S\times V}}\leq 2^{-\frac{1}{2}\left(H_{min}\left(X\right)-\log_{2}\left|V\right|\right)}.$$

**Corollary** **1.**
*Under the same assumptions, we obtain that*

$$\begin{array}{cc} H_{min}\left(D_{f_{Y}\left(X\right)}\right)\; & \geq -\log_{2}\left(\frac{1}{\left|V\right|}+2^{-\frac{1}{2}\left(H_{min}\left(X\right)-\log_{2}\left|V\right|\right)+1}\right)\\ \; & \approx \log_{2}\left|V\right|-\frac{\left|V\right|}{\ln2}2^{-\frac{1}{2}\left(H_{min}\left(X\right)-\log_{2}\left|V\right|\right)+1}\\ \; & =\log_{2}\left|V\right|-\epsilon . \end{array}$$



**Proof.** Let $U_{V}$ be the uniform distribution on *V*. By triangle inequality and Leftover Hash Lemma,
$$\begin{array}{cc} H_{min}\left(D_{f_{Y}\left(X\right)}\right) & =-\log_{2}{∥D_{f_{Y}\left(X\right)}∥}_{\infty }\\ \; & \geq -\log_{2}\left[{∥U_{V}∥}_{\infty }+{∥D_{f_{Y}\left(X\right)}-U_{V}∥}_{\infty }\right]\\ \; & \geq -\log_{2}\left[{∥U_{V}∥}_{\infty }+{∥D_{f_{Y}\left(X\right)}-U_{V}∥}_{1}\right]\\ \; & \geq -\log_{2}\left[{∥U_{V}∥}_{\infty }+{∥D_{\left(Y,f_{Y}\left(X\right)\right)}-U_{S\times V}∥}_{1}\right]\\ \; & =-\log_{2}\left[{∥U_{V}∥}_{\infty }+2\Delta_{D_{\left(Y,f_{Y}\left(X\right)\right)},U_{S\times V}}\right]\\ \; & \geq -\log_{2}\left(\frac{1}{\left|V\right|}+2^{-\frac{1}{2}\left(H_{min}\left(X\right)-\log_{2}\left|V\right|\right)+1}\right)\\ \; & \approx \log_{2}\left|V\right|-\frac{\left|V\right|}{\ln2}2^{-\frac{1}{2}\left(H_{min}\left(X\right)-\log_{2}\left|V\right|\right)+1}. \end{array}$$The last approximation is due to Taylor expansion at $x=\frac{1}{\left|V\right|}$. □

The Leftover Hash Lemma is effective in generating high-quality output sequences in the following situations:(i)      When $H_{min}\left(X\right)$ is smaller than |T|.(ii)     When $H_{min}\left(X\right)$ is significantly larger than |V|.(iii)    When |S| is substantially smaller than |V|.

The key point here is the last condition. To generate output sequences using the Leftover Hash Lemma, a hash function must be uniformly selected from the 2-universal hash family. This means that random numbers are required to generate random numbers. If the size of the 2-universal hash family is small, a large amount of random numbers can be generated using a small amount of random numbers. Therefore, constructing a 2-universal hash family of small size is the key point in the use of the Leftover Hash Lemma.

**Example** **1.**Suppose that $T={\left\{0,1\right\}}^{900}$, $V={\left\{0,1\right\}}^{100}$, $\left|S\right|=2^{30}$, $H_{min}\left(X\right)=550$, $H_{min}\left(Y\right)=30$, then by Corollary 1, $H_{min}\left(D_{f_{Y}\left(X\right)}\right)\geq 100-\frac{1}{\ln2}2^{-124}$. This implies that if we have a low quality entropy source with a min-entropy of 550 out of 900, and a 2-universal hash family of size $2^{30}$, we can leverage a 30-bit random number generator to produce nearly random 100-bit numbers.

#### 4.1.2. Slow-Refresh of IDQ QRNG [19,20]

IDQ QRNG uses an m×n random matrix as the entropy accumulation function. This function transforms an input sequence of length *m* into an output sequence of length *n*. It can also be readily proven that this collection of matrices forms a 2-universal hash family [4]. Using the notation from the Leftover Hash Lemma, this can be represented as
$$T={\left\{0,1\right\}}^{n},\;V={\left\{0,1\right\}}^{m},\;S={\left\{0,1\right\}}^{m\times n}.$$

The min-entropy per bit of the quantum entropy source (which corresponds to $H_{min}\left(X\right)$ in our notation) is not disclosed. However, according to [19], the ϵ value is designed to be set to $2^{-100}$. The random matrix is generated only once, and the mn elements that make up this matrix are generated using a 1-bit random number generator mn times. The 1-bit random number generator creates a single bit by collecting multiple bits from the digitized entropy source and performing an XOR operation. The more bits that are XORed, the higher the min-entropy of the generated 1-bit. This can be confirmed through the Piling Up Lemma. Figure 11 illustrates the entropy accumulation of IDQ QRNG.

#### 4.1.3. Limitations of IDQ QRNG Entropy Accumulation Model

At first glance, the entropy accumulation model of IDQ QRNG seems to generate random numbers based on the Leftover Hash Lemma, but this model has two inherent limitations.

The first point concerns the safety of the 1-bit random number generator used to create the random matrix. This generator should output 0 and 1 with a probability of 1/2. However, the min-entropy of the entropy source used in IDQ QRNG and the iteration number of XOR operations are not specified, which makes it impossible to confirm whether the random matrix was uniformly generated.

The second point is that the random matrix is generated just one time in IDQ QRNG, but this implementation seems an incomplete application of the Leftover Hash Lemma. If the random matrix is continuously used, the condition of independence between the consecutive *m*-bit output sequences is compromised. Thus, in this case, it is impossible to obtain the overall entropy of long output sequence by the Leftover Hash Lemma. For example, if we generate *m*-bit output sequences $X_{i}$’s (1≤i≤5) by using the entropy accumulation of IDQ QRNG, then although for each 1≤i≤5, $H_{min}\left(X_{i}\right)=m-\epsilon$, it does not always satisfy that $H_{min}(X_{1}∥X_{2}∥X_{3}∥X_{4}∥X_{5})=5\left(m-\epsilon \right)$. Therefore, the matrix should be updated whenever a new output sequence is generated in order to properly apply the Leftover Hash Lemma.

#### 4.1.4. Differences between IDQ QRNG and Our Entropy Accumulation

The simple characteristic distinguishing Fast-Refresh from Slow-Refresh is the length of the input sequence. In our entropy accumulation, we generate the output sequence by XORing five 8-bit input sequences, that is, the total length of the input sequence is 40-bit with the 8-bit length output sequence. However, IDQ QRNG uses the input sequence of 1024-bit or 2048-bit, and the output sequences of 768-bit or 1792-bit, respectively [19]. The reason for such a long length of the input sequence is to adjust the ϵ of the Leftover Hash Lemma to about $2^{-100}$ [19].

The bit loss rate is another characteristic distinguishing Fast-Refresh from Slow-Refresh. In our entropy accumulation process, 32-bit are discarded from 40-bit of input sequence, resulting in a bit loss rate of 80%. On the other hand, in the IDQ QRNG entropy accumulation process, 256-bit are discarded from 1024-bit or 2048-bit of input sequence, resulting in bit loss rates of 25% or 12.5%, respectively. Through this, we can see that the Slow-Refresh handles a large number of bits at once but has a relatively low loss rate. However, a low bit loss rate results in a decrease in operation speed. The major differences are shown in Table 2.

### 4.2. Comparing with the Fast-Refresh of Windows RNG

We have already described how Fast-Refresh of Windows RNG works in the introduction. Hence, we describe the entropy accumulation theory in [2] and calculate the number of operations that must be iterated when applying it to the input sequence $Y_{j}$. First, we describe the 2-monotone distribution and the covering number, which are essential concepts in [2]. Thereafter, we present Theorem 5.2 of [2] (which is the main result of [2]) with our notation.

Next, we explain why [2] cannot be directly applied to our entropy accumulation model and suggest an additional S-box operation as a solution. With this additional operation, we can apply Theorem 5.2 of [2] and overcome the limitations of the original theory. Finally, we provide the theoretical number of operations necessary to guarantee a min-entropy of 7.86 per 8-bit when Theorem 5.2 of [2] is applied to the RNG.

#### 4.2.1. Main Results of [2]

The covering number is used to measure the efficiency of the permutations used in entropy accumulation. The efficiency of entropy accumulation increased as the covering number of permutations decreased.

**Definition** **5**(Covering number)**.** *For a permutation $\pi :\left\{0,1,...,n-1\right\}\to \left\{0,1,...,n-1\right\}$, and an integer 1≤k≤n, the coveringnumber $C_{\pi ,k}$ is the smallest natural number m such that*
(8)$$\{\pi^{l}\left(j\right):0\leq j<k,\;0\leq l<m\}=\left\{0,1,...,n-1\right\}.$$*If no such m exists, then $C_{\pi ,k}=\infty$;*

Figure 12 shows the calculation of the number of coverings.

One special property of an entropy source in [2] is 2-monotone distribution. The definition is as follows:

**Definition** **6**(2-monotone distribution)**.** *The probability distribution $D_{X}$ of an n-bit random variable X follows a 2-monotone distribution if its domain can be divided into two disjoint intervals, where $D_{X}$ is a monotone function.*

A 2-monotone distribution has at least one inflection point (peak), because it is divided into two monotonic intervals.

Based on these definitions, we can state Theorem 5.2 of [2].

**Theorem** **2**(Theorem 5.2 of [2])**.** *Suppose that for independent n-bit random variables $Y_{1},Y_{2},...,Y_{l}$, the probability distributions $D_{Y_{1}},D_{Y_{2}},...,D_{Y_{l}}$ have a min-entropy of at least k(≥2) and follow a 2-monotone distribution. Let $\pi :\left\{0,1,...,n-1\right\}\to \left\{0,1,...,n-1\right\}$ be a permutation and $m=C_{\pi ,k^{\prime }}$ be a covering number where $k^{\prime }=\left\lfloor \frac{k}{2}\right\rfloor$. Let $\Lambda_{\pi }^{\left(l\right)}=A_{\pi }^{l-1}\left(Y_{1}\right)⊕A_{\pi }^{l-2}\left(Y_{2}\right)⊕\cdots ⊕Y_{l}$. Then, for any l≥m,*
$$H_{min}\left(D_{\Lambda_{\pi }^{\left(l\right)}}\right)\geq n-\left(\left\lfloor \frac{n}{k^{\prime }}\right\rfloor +1\right)\cdot log_{2}\left(1+2^{k^{\prime }-\left(\frac{k}{2}\right)\left\lfloor \frac{l}{m}\right\rfloor }\right)\approx n\left(1-2^{\frac{k}{2}-\frac{kl}{2m}}\right).$$

One can easily observe that as *n* increased, $H_{min}\left(D_{\Lambda_{\pi }^{\left(l\right)}}\right)$ converged to *n*.

#### 4.2.2. Windows RNG Entropy Accumulation without 2-Monotone Condition

Theorem 2 provides a min-entropy lower bound for $\Lambda_{\pi }^{\left(l\right)}$ with only three restrictions. The conditions under which the input sequences are independent, and the covering number of permutations is finite, are relatively easy to satisfy. However, it is challenging to satisfy the condition that all input sequences follow a 2-monotone distribution. In particular, for the image sensor entropy sources used, input sequences that follow a 2-monotone distribution are unattainable.

We generated $Y_{j}$ by concatenating four 2-bit entropy sources: $W_{4j-3},W_{4j-2},W_{4j-1},W_{4j}$. We used this method only for experimental verification (SP-800-90b requires at least 8-bit data), and $H_{min}\left(\Gamma^{\left(l\right)}\right)$ was theoretically calculated by adding the four min-entropy values of the 2-bit segment of $\Gamma^{\left(l\right)}$ (see Section 3.4). However, if we apply Theorem 2 to our input sequences, we cannot use the divide-and-conquer approach. This is because while $D_{W_{i}}$ definitely satisfies a 2-monotone (as can be observed in Figure 10), Theorem 2 requires input sequences with a min-entropy of two or more. Note that the 2-bit entropy sources $W_{i}$ cannot achieve this condition. For this reason, when applying Theorem 2 to our input sequences, we must use the concatenation method for theoretical, rather than experimental, reasons. However, if the input sequences are processed in a concatenated manner, it is impossible to satisfy the 2-monotone assumption. We explain this using a 4-bit example. Figure 13 represents the probability distribution of 2-bit entropy sources $W_{1}$ and $W_{2}$ that follow a 2-monotone distribution. The probability distribution of $W_{1}∥W_{2}$, created by concatenating the two entropy sources, is shown in Figure 14.

If we consider {4i−3,4i−2,4i−1,4i} as one group, there are four groups and the overall shape of $D_{W_{1}∥W_{2}}$ is similar to that of $D_{W_{1}}$. However, the shape of each group’s graph is similar to $D_{W_{2}}$. The shape of such a concatenated distribution tends to become more complex as the entropy sources become more concatenated. Therefore, inevitably, the distribution of the input sequences assumes a shape that is far from a 2-monotone. However, if $Y_{j}$ passes through a “good” S-box, the data can be transformed to follow a 2-monotone distribution (in particular, a monotone distribution). For example, if $W_{1}∥W_{2}$ passes through an S-box *S* in Table 3, $D_{S(W_{1}∥W_{2})}$ follows a monotone distribution. This is illustrated in Figure 15. The method for creating such *S* is simple. After obtaining the distribution of concatenated data, arrange the distribution values in ascending order and input the elements of the domain that provide the distribution values into the S-box in order. For example, as shown in Figure 14, $D_{W_{1}∥W_{2}}$ has a minimum value at x=15 and the second-smallest value at x=3. If *x* values are arranged in this manner, they become 15, 3, 12, 7,⋯, 2, 6, and 10. Inputting these in sequence into the S-box creates *S* results in Table 3.

Although we provide an example of transforming 4-bit data created by concatenating two 2-bit entropy sources, this method can be applied to arbitrary *n*-bit data. With this method, even if the input sequences do not follow the 2-monotone distribution, Theorem 2 can be applied. However, memory is required to store the S-box, and the accumulation speed can be reduced owing to additional operations. Additionally, significant amounts of meaningful data may be required to estimate the distribution. Figure 16 illustrates the Windows RNG entropy accumulation process using an additional S-box.

#### 4.2.3. Applying Theorem 2

In this subsection, we determine the number *l* required for $\Lambda_{\pi }^{\left(l\right)}$ to exceed a min-entropy of 7.86-bit per 8-bit by applying Theorem 2 to $S(Y_{j})$. We did not conduct actual experiments because of the numerous S-boxes that needed to be implemented and the vast amount of data required to estimate $D_{Y_{j}}$.

To apply Theorem 2, the first step is obtaining $H_{min}\left(D_{S(Y_{j})}\right)$; as the S-box does not change the min-entropy, $H_{min}\left(D_{Y_{j}}\right)$ is used instead. Using [17] to estimate $H_{min}\left(D_{Y_{j}}\right)$ requires 1,000,000 pieces of $Y_{j}$, which is practically impossible. Therefore, we set k=4.9378, which is the lower bound of $H_{min}\left(D_{Y_{j}}\right)$, estimated using 2000 pieces of $W_{i}$ data, as mentioned in Section 3.5. Note that the estimated min-entropy at l=1 in Table 1 of Section 3.5 and the previously estimated min-entropy have explicitly different estimation targets. With k=4.9378, $k^{\prime }=\left\lfloor \frac{k}{2}\right\rfloor =2$. If we set $\pi =rot_{\left(2,8\right)}$, the covering number $m=C_{\pi ,k’}$ becomes four, which is the minimum value of every possible π. Thus, we opted to use $rot_{\left(2,8\right)}$ as a bit permutation for the entropy accumulation. Considering these considerations, *l* can be calculated as follows:$$\begin{array}{cc} \; & n\left(1-2^{\frac{k}{2}-\frac{kl}{2m}}\right)\geq \;7.86\\ \; & \Rightarrow \frac{n-7.86}{n}\geq 2^{\frac{k}{2}\left(1-\frac{l}{m}\right)}\\ \; & \Rightarrow \frac{\ln\left(n-7.86\right)-\ln n}{\ln2}\geq \frac{k}{2}\left(1-\frac{l}{m}\right)\\ \; & \Rightarrow l\geq m\left[1-\frac{2}{k}\left(\frac{\ln\left(n-7.86\right)-\ln n}{\ln2}\right)\right]\\ \; & \Rightarrow l\geq 4\left[1-\frac{2}{4.9378}\left(\frac{\ln\left(8-7.86\right)-\ln8}{\ln2}\right)\right]=13.4559. \end{array}$$

That is, if we assume l=14 to create $\Lambda_{\pi }^{\left(l\right)}$, $H_{min}\left(D_{\Lambda_{\pi }^{\left(l\right)}}\right)$ will exceed the value 7.86.

In Section 3.4, we observed that our theory yields l=16. Although Theorem 2 may yield slightly superior results, the difference is insignificant. Considering the time consumed for the additional S-box and bit permutations and the storage space for S-boxes, we can conclude that our entropy accumulation provides more practical results.

## 5. Conclusions

The contributions of our study can be summarized as follows. First, we have proposed entropy accumulation of the Fast-Refresh type, which is composed of bitwise XOR alone without hash functions, and have proved the theorem that requires only the independence without identical distribution condition of input sequences.

Second, we have established 7.86 as the lower bound for the min-entropy per 8-bit, which was considered secure based on the three benchmarks. To surpass this lower bound, our proposed theory yielded iteration number l=16.

We have implemented an actual RNG to verify the theory. Our experimental results have indicated that if we use XOR operations just four times, the generated output sequences exceeded the lower bound. The entropy source used in this experiment is an image sensor PV 4209 K. This entropy source is a QRNG that utilizes dark shot noise to generate random numbers. The most important property of our entropy source is the independence of pixels. Since each piece of 2-bit data from pixels is considered as an independent random variable, we can apply the main theorem to obtain the lower bound of the min-entropy.

Finally, we have compared our entropy accumulation with two types of entropy accumulations, which are Slow-Refresh of IDQ QRNG and Fast-Refresh of Windows RNG.

As a further study, we would like to consider various entropy accumulations that have more general and practical applications than our proposed Fast-Refresh mechanism.

## Figures and Tables

**Figure 1 entropy-25-01056-f001:**
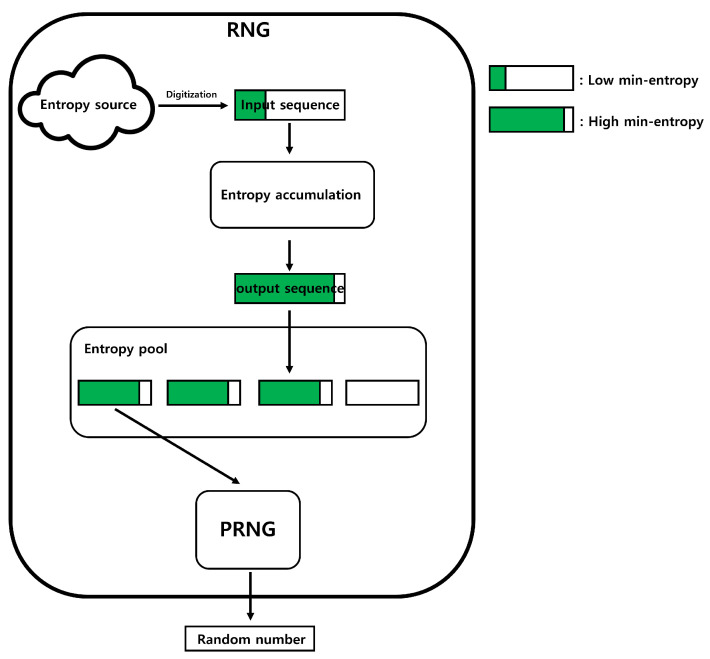
Operation of an RNG.

**Figure 2 entropy-25-01056-f002:**
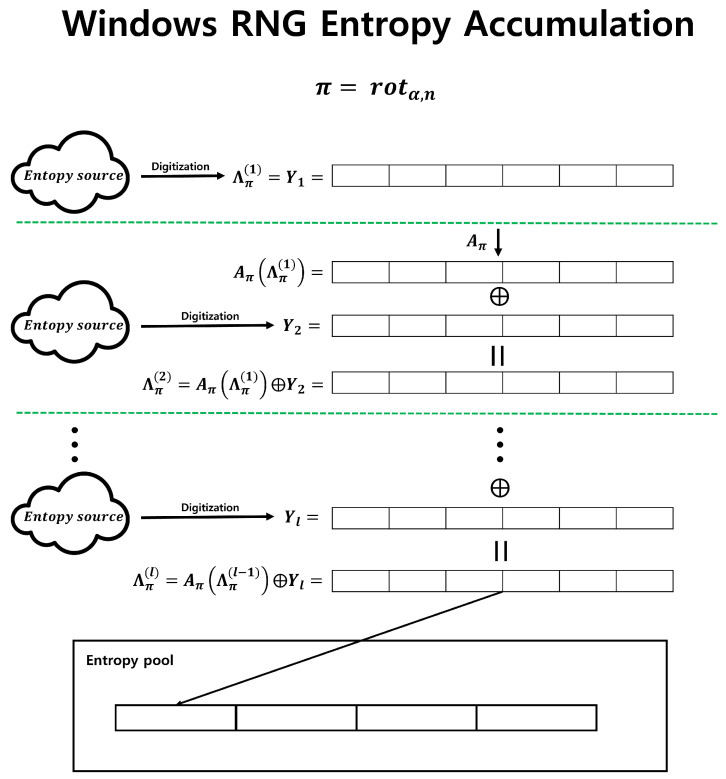
Windows RNG entropy accumulation.

**Figure 3 entropy-25-01056-f003:**
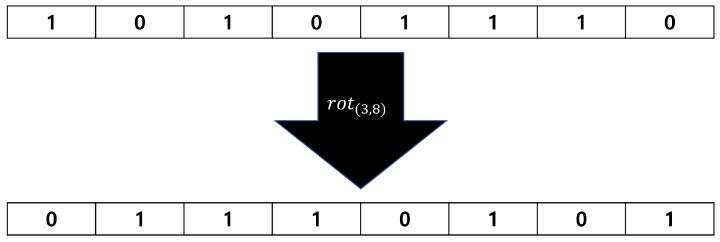
Example of the rot permutation when n=8 and α=3.

**Figure 4 entropy-25-01056-f004:**
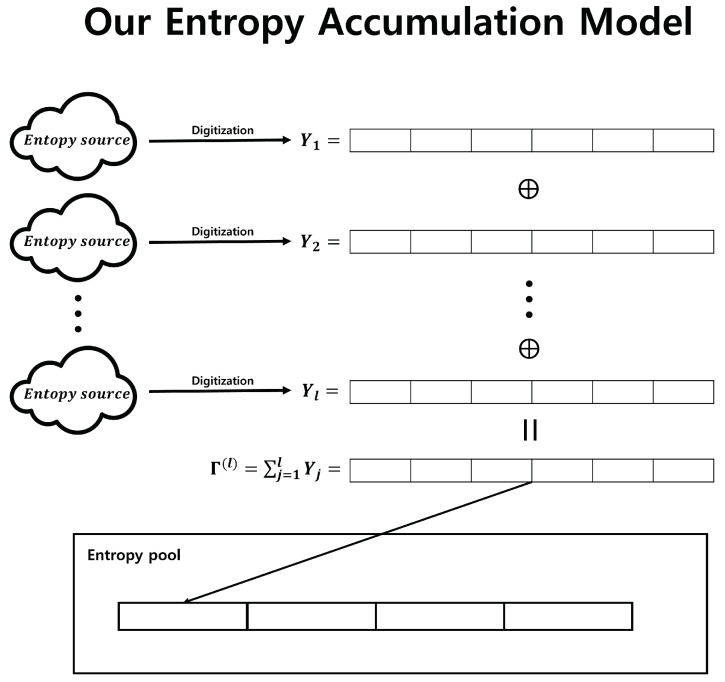
Entropy accumulation using only bitwise XOR operations.

**Figure 5 entropy-25-01056-f005:**
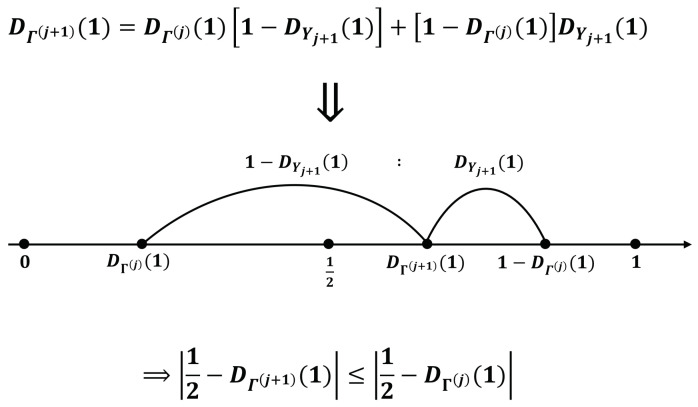
Our entropy accumulation with n=1.

**Figure 6 entropy-25-01056-f006:**
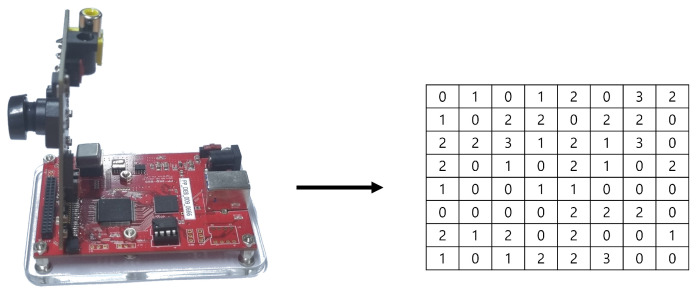
Data transmission process of image sensor.

**Figure 7 entropy-25-01056-f007:**
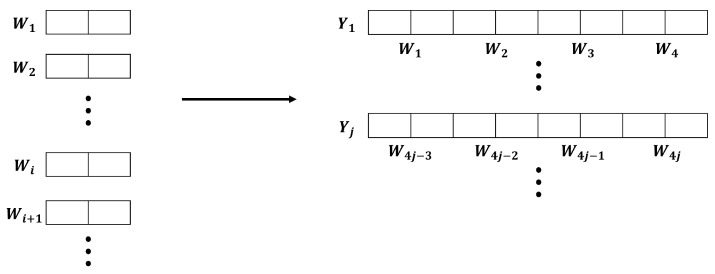
Concatenation of 2-bit data to form 8-bit input sequence.

**Figure 8 entropy-25-01056-f008:**
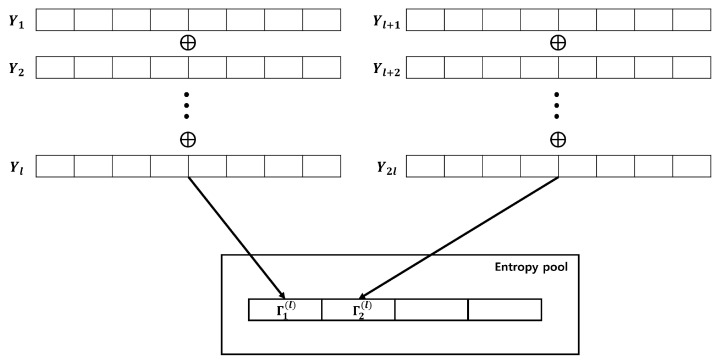
Accumulating entropy source using XOR operation.

**Figure 9 entropy-25-01056-f009:**
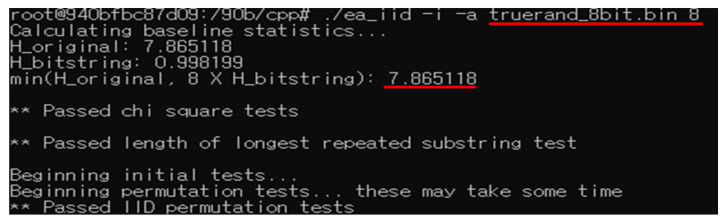
Min-entropy of true random 8-bit.

**Figure 10 entropy-25-01056-f010:**
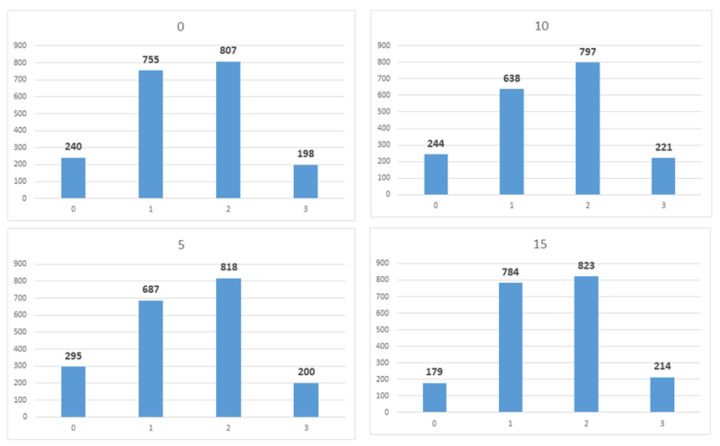
Probability distribution of each OBP.

**Figure 11 entropy-25-01056-f011:**
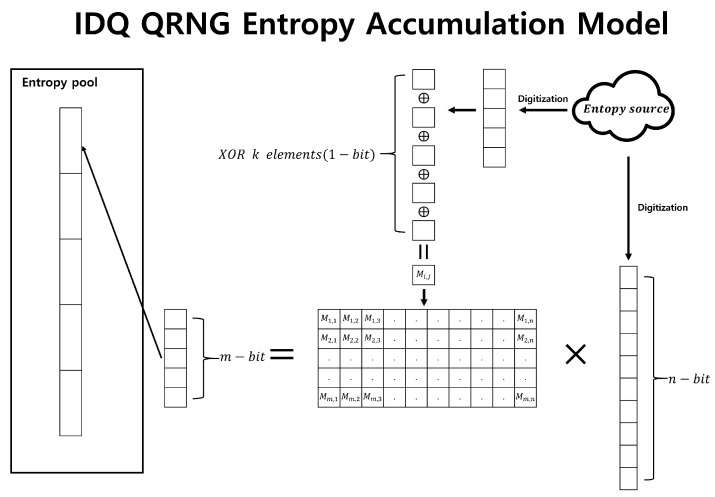
IDQ QRNG entropy accumulation model.

**Figure 12 entropy-25-01056-f012:**
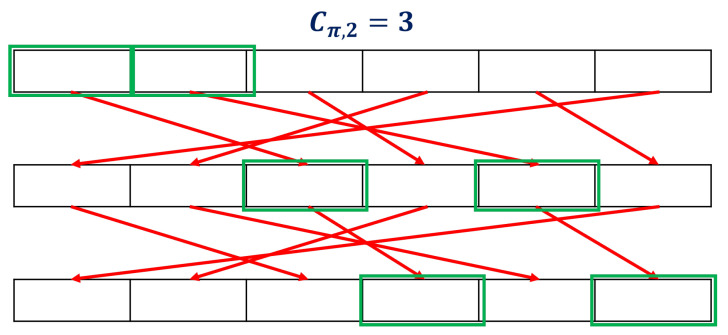
Illustration of covering number when $C_{\pi ,2}=3$. The covering number is 3 because all bits are covered using the permutation operation twice.

**Figure 13 entropy-25-01056-f013:**
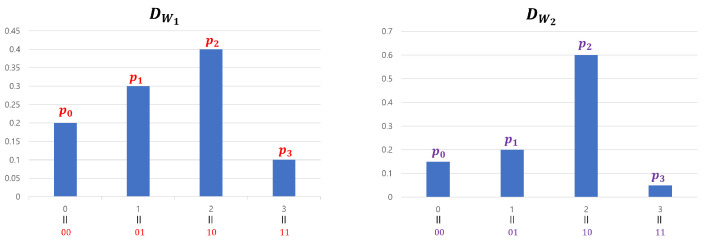
Example of two distributions that follow a 2-monotone distribution.

**Figure 14 entropy-25-01056-f014:**
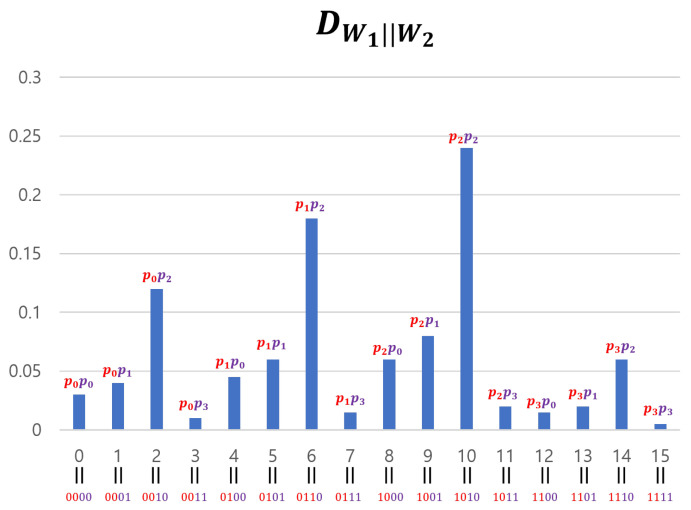
Distribution of concatenated entropy sources.

**Figure 15 entropy-25-01056-f015:**
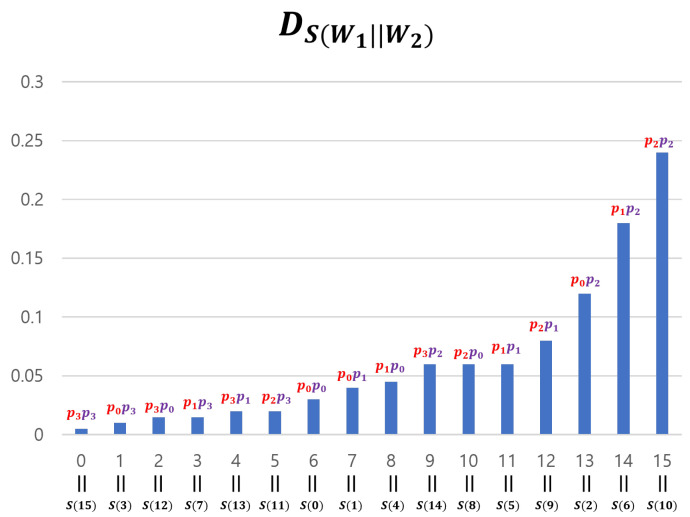
Distribution of S-boxed concatenated entropy sources.

**Figure 16 entropy-25-01056-f016:**
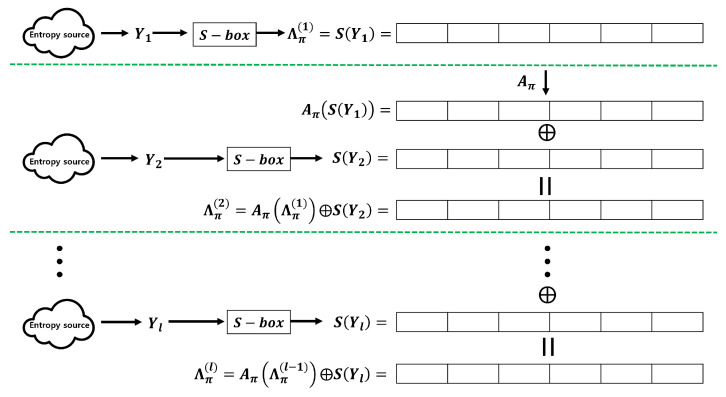
Windows RNG entropy accumulation with additional S-box.

**Table 1 entropy-25-01056-t001:** Min-entropy values corresponding to number of XOR operation repetitions.

*l*	*i*	*j*	*k*	Min-Entropy per 8-Bit
1	4,000,000	1,000,000	1,000,000	3.305
2	8,000,000	2,000,000	1,000,000	7.115
3	12,000,000	3,000,000	1,000,000	7.700
4	16,000,000	4,000,000	1,000,000	7.852
5	20,000,000	5,000,000	1,000,000	7.864

**Table 2 entropy-25-01056-t002:** Major differences between our entropy accumulation and that of IDQ QRNG.

	Our Entropy Accumulation	IDQ QRNG
Refresh Type	Fast-Refresh	Slow-Refresh
Theoretical Background	Fourier Transform	Leftover Hash Lemma
Implementation Aspect	Simple XOR operations	Difficulty of implementing universal hash family
Input Sequence Length	40	1024 or 2048
Output Sequence Length	8	768 or 1792
Bit Loss Rate	80%	25% or 12.5%

**Table 3 entropy-25-01056-t003:** 4-bit S-box, which is used with $W_{1}∥W_{2}$ to create monotone distribution.

*x*	15	3	12	7	13	11	0	1	4	14	8	5	9	2	6	10
S(x)	0	1	2	3	4	5	6	7	8	9	10	11	12	13	14	15

## Data Availability

The data presented in this study are available on request from the corresponding author.

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
