# Peer review of "Practical Entropy Accumulation for Random Number Generators with Image Sensor-Based Quantum Noise Sources"

_entropy, 2023, doi:10.3390/e25071056_

Round 1
Reviewer 1 Report
The paper titled "Practical Entropy Accumulation for Random Number Generators with Image Sensor-Based Quantum Noise Sources" demonstrates significant potential and warrants acceptance after major revision. The paper proposes a new and efficient method for entropy accumulation in random number generators, utilizing bitwise XOR operations. The authors validate their approach by evaluating it on a quantum random number generator that utilizes dark noise from image sensor pixels as an entropy source.
The introduction of the paper effectively outlines the three key processes involved in a random number generator: digitization, entropy accumulation, and pseudorandom number generation (PRNG). The authors emphasize the significance of digitization, which involves converting entropy sources into binary data known as the input sequence. However, they highlight the common issue of low min-entropy in the input sequence.
The proposed method for entropy accumulation using bitwise XOR operations is a notable contribution of this paper. By applying this approach, the authors determine the theoretical bound for the min-entropy of the output random sequence. The experimental results align with these theoretical bounds, demonstrating that the achieved min-entropy surpasses the threshold of 7.86 per 8 bits, even with just four XOR operations. The corresponding results, including the min-entropy per 8-bit for various values, are thoughtfully presented in Table 1.
In conclusion, the paper proposes a practical and efficient solution for entropy accumulation in random number generators. The authors provide valuable contributions by proposing the new method, establishing theoretical bounds, validating their approach through experiments, and offering insights into the quantum random number generator utilizing image sensor-based quantum noise sources. However, some limitations should be addressed. The paper primarily evaluates the proposed method on a specific entropy source, necessitating further investigation into its generalizability to other sources. Additionally, a more comprehensive comparison with existing entropy accumulation methods regarding performance and efficiency would enhance the paper's contributions.
-
Reviewer 2 Report
High quality random number is essential in both clasiscal quantum cryptography. In this work, the authors derived a theoretical bound for the min-entropy of the output random sequence through entropy accumulation using only bitwise XOR operations of independent input sequences of entropy source. They examined the results on quantum random number generator that uses dark noise of image sensor pixels. They established 7.86 as the lower bound for the min-entropy per 8-bits. The experimental results indicated 4 times of use of XOR operations generated output sequences exceeded the lower bound, showing the feasibility of their scheme.
I recommend its acceptance after revision. In the revision, the two related references of RNG using noises are suggested [r1,r2], one uses the the tunneling effect and one with vaccuum fluctuation. It is worth pointing that use of random numbers in random matrix production calculation, quantum key distribution and even topological quantum code [r3,r4,r5].
[r1] Zhou H, Li J, Zhang W, et al. Quantum random-number generator based on tunneling effects in a Si diode. Physical Review Applied, 2019, 11(3): 034060.
[r2] Zhou Q, Valivarthi R, John C, et al. Practical quantum random‐number generation based on sampling vacuum fluctuations. Quantum Engineering, 2019, 1(1): e8.
[r3] Bao N, Lu J, Cai R, et al. Computing growth rates of random matrix products via generating functions. AAPPS Bulletin, 2022, 32(1): 28.
[r4] Kwek L C, Cao L, Luo W, et al. Chip-based quantum key distribution. AAPPS Bulletin, 2021, 31: 15
[r5] Li Ding, Haowen Wang, Yinuo Wang, Shumei Wang, "Based on Quantum Topological Stabilizer Color Code Morphism Neural Network Decoder", Quantum Engineering, vol. 2022, Article ID 9638108, 2022.
Reviewer 3 Report
The manuscript develops the technologies of random number generators (RNG). A random number generator that uses only bitwise XOR operations (which is important for the speed) and quantum noise (entropy) source is proposed, min-entropy of this source was estimated and theorems about the lower bound for the min-entropy of the final sequence are proved. I believe that the results are interesting and worth publishing. The text is well-written.
I have one question about the research and results and one comment about the text. The question about the results is as follows. The authors stress that their RNG does not use hash functions and uses only bitwise XOR. However, the well-known hash function based on the multiplication with the Toeplitz binary matrix also uses only bitwise XOR. It is necessary to comment why the proposed RNG is better (or not?) than the Toeplitz hashing-based ones.
A comments about the text is as follows: I think that it is worthwhile to mention the role of a quantum noise source in conclusions since this is one of the novelties of the manuscript.
Round 2
Reviewer 1 Report
The Authors reviewed all the issues presented by the reviewers, and the paper can be accepted in its present form
The Authors reviewed all the issues presented by the reviewers and the paper can be accepted in its present form